# Pupillometry Reveals the Role of Arousal in a Postexercise Benefit to Executive Function

**DOI:** 10.3390/brainsci11081048

**Published:** 2021-08-07

**Authors:** Naila Ayala, Matthew Heath

**Affiliations:** 1Department of Kinesiology, School of Kinesiology, University of Western Ontario, London, ON N6G 3K7, Canada; mheath2@uwo.ca; 2Graduate Program in Neuroscience, University of Western Ontario, London, ON N6G 3K7, Canada; 3Department of Kinesiology, University of Waterloo, Waterloo, ON N2L 3G1, Canada

**Keywords:** antisaccade, cognition, exercise, oculomotor, task-evoked pupil dilation, vision

## Abstract

A single bout of aerobic exercise improves executive function; however, the mechanism(s) underlying this improvement remains unclear. Here, we employed a 20-min bout of aerobic exercise, and at pre- and immediate post-exercise sessions examined executive function via pro- (i.e., saccade to veridical target location) and anti-saccade (i.e., saccade mirror symmetrical to a target) performance and pupillometry metrics. Notably, tonic and phasic pupillometry responses in oculomotor control provided a framework to determine the degree that arousal and/or executive resource recruitment influence behavior. Results demonstrated a pre- to post-exercise decrease in pro- and anti-saccade reaction times (*p* = 0.01) concurrent with a decrease and increase in tonic baseline pupil size and task-evoked pupil dilations, respectively (ps < 0.03). Such results demonstrate that an exercise-induced improvement in saccade performance is related to an executive-mediated “shift” in physiological and/or psychological arousal, supported by the locus coeruleus norepinephrine system to optimize task engagement.

## 1. Introduction

Top-down executive control includes response suppression, working memory, and cognitive flexibility, and each component is essential for daily living [1]. A single bout of aerobic and/or resistance training improves executive function [2,3] and is a benefit that persists for up to 60 min [4]. A prominent mechanism associated with the benefit is an exercise-mediated change in arousal [5]. Notably, arousal is a multidimensional construct (i.e., physiological, cognitive/psychological, and affective components) [5], and little research has examined the effect of exercise on the distinct factors clustered within the term. A crucial component of cognitive/psychological arousal is the locus coeruleus norepinephrine (LC-NE) system, which is a collection of noradrenergic neurons within the brainstem that have an essential role in modulating the neural system’s level of alertness and the brain’s attentional state [6]. Therefore, the present investigation sought to determine whether a single bout of aerobic exercise influences LC-NE activity and how a putative change may influence a post exercise benefit to executive function. 

The LC-NE system influences sensory processing and cognition through the regulation of attention [7]. Accordingly, behavioral and neuropsychological measures that reflect the LC-NE system provide a framework to understand the mechanism(s) supporting the post exercise improvement in executive function. The present work used concurrent behavioral and pupillometry metrics of pro- (i.e., saccade to veridical target location) and anti-saccades (i.e., saccade mirror symmetrical to a target stimulus) to examine pre- and immediate post-exercise executive function. The basis for this was three-fold. First, behavioral and electrophysiological studies have shown that antisaccades are mediated via an extensive frontoparietal network [8] that shows task-dependent changes in activity following single and chronic bouts of exercise [9,10]. Second, antisaccades provide the necessary resolution to detect subtle post exercise improvements in executive function (i.e., decreased reaction times (RT)) across a range of exercise intensities and durations in healthy young and older adults, as well as those at risk for cognitive decline [11,12]. Third, the pupil size changes observed in the antisaccade task provide a proxy for task-dependent changes in executive function (i.e., inhibitory control) and cognitive/psychological arousal [7,13,14]. Indeed, the locus coeruleus (LC) receives direct—and indirect—input from the prefrontal cortex and the insula, which in turn influence the efferent gain throughout the cortical and subcortical regions, serving sensorimotor and cognitive processing [15,16]. These projections influence a biphasic pupil response that are indicative of arousal and executive resource recruitment: a tonic (baseline pupil size) and a phasic (task-evoked pupil dilation: TEPD) response [7,17]. During tasks that require a focusing of attention, neurons in the LC exhibit moderate levels of tonic activation, which enable phasic bursts of activity to occur that support the execution of a response to task-related events [7]. Arousal modulates baseline pupil size in a manner resembling the classic Yerkes−Dodson inverted-U relationship, and this influences TEPD and cognitive performance in a stereotypical manner [17]. Previous work has demonstrated that increasing tonic activation decreases phasic bursts, increases distractibility, and decreases performance (i.e., an increase in RT and/or response errors), whereas suppressing tonic activation to moderate levels increases phasic bursts, reduces distractibility, and improves performance (i.e., a decrease in RT and/or response errors) [7,17]. Consequently, if tonic activation falls below this moderate level, then task-relevant information is not processed and reflects a low level of alertness in the neural system [6,7].

Here, we examined antisaccade performance and pupillometry metrics prior to and following a 20-min single-bout of aerobic activity via a cycle ergometer at 80% of the participants’ predicted maximum heart rate (HR_max_: 220 minus age in years). In terms of research predictions, if exercise enhances LC-NE system attentional modulation via a reduction in cognitive/psychological arousal, then decreased post exercise antisaccade RTs should be paired with a pre- to post-exercise decrease in baseline pupil size (i.e., suppression of tonic activity to moderate levels) and a concomitant increase in TEPD (i.e., increased task-evoked phasic bursts). Such a pattern of results would evince improved executive control via a narrowing of selective attention and serve to optimize the processing of task-relevant information. In contrast, if a post exercise improvement in antisaccade planning is executive-specific and not related to a modulation of attentional control via cognitive/psychological arousal, then decreased post exercise antisaccade RTs should be paired with a post exercise increase in antisaccade TEPDs and no change in baseline pupil size.

## 2. Materials and Methods

### 2.1. Participants

Sixteen (eight females in the age range of 20–26 years) members of the University of Western Ontario community participated in this study. All were self-declared right-hand dominant, had normal or corrected-to-normal vision, and no current or previous history of neuropsychiatric or neurological impairment. Participants obtained a full score on the Physical Activity Readiness Questionnaire (PAR-Q) and were “recreationally active” as determined by the Godin Leisure-Time Exercise Questionnaire (GLTEQ; mean = 63, SD = 16, min = 32, and max = 88). Participants refrained from caffeine and tobacco use 8 h prior to participation. Participants signed a consent form approved by the Health Sciences Research Ethics Board, University of Western Ontario, and this research was conducted according to the Declaration of Helsinki. 

### 2.2. Exercise Intervention

The exercise intervention involved a 20-min bout of aerobic exercise via a cycle ergometer (Monark 818E Ergometer, Monark Exercise AB, Vonsbro, Sweden) at 80% of the participants’ HR_max_. Prior to and after the intervention, a 2.5 min warm-up and cool down, respectively, were performed at 50% of HR_max_ (Heath and Shukla, 2020). Heart rate was continuously monitored during the intervention (Polar Wearlink and Coded Transmitter, Polar Electro Inc., Lack Success, NY, USA), and the experimenter or participant adjusted ergometer resistance to maintain a work rate in the prescribed intensity.

### 2.3. Oculomotor Task

Prior to and following the exercise session, participants sat on a height adjustable chair in front of a table with their head placed in a head−chin rest. A 30-in LCD monitor (60 Hz, 8 ms response rate, 1280 × 960 pixels; Dell 3007WFP, Round Rock, TX, USA) was located at the participants’ midline and 550 mm from the front edge of the tabletop and was used to present visual stimuli. Gaze position and pupil size of the left eye were sampled at 1000 Hz (EyeLink 1000 Plus; SR Research Ltd., Ottawa, ON, Canada). Stimulus presentation and data acquisition were controlled via MATLAB (R2018b, TheMathWorks, Natick, MA, USA) and the Psychophysics Toolbox extensions (v. 3.0) [18], including the EyeLink Toolbox [19]. Prior to data collection, a nine-point calibration was performed and followed by a validation (i.e., <1° of error). 

Visual stimuli were presented on a high-contrast black background (1 cd/m^2^) and included a centrally presented red or green fixation cross (1°). The color of the fixation was equiluminant (42 cd/m^2^) and instructed the nature of the required response (i.e., prosaccade = green and antisaccade = red). Open white circles served as targets (2.7° diameter: 132 cd/m^2^) and were 13.5° (i.e., proximal) and 16.5° (i.e., distal) left and right of the fixation, respectively, and in the same horizontal axis. The different eccentricities were used to prevent participants from adopting stereotyped responses. A trial began with the appearance of the fixation for 1000 ms, after which it was extinguished and a target appeared 200 ms thereafter (i.e., gap paradigm). Targets were presented for 50 ms and this brief presentation—in part –served to equate pro- and anti-saccades for the absence of extraretinal feedback [20]. Target onset cued participants to pro- (i.e., saccade to veridical target location) or anti-saccade (i.e., saccade mirror symmetrical to target location) “quickly and accurately”. Pro- and anti-saccades, as well as stimulus location (i.e., left and right of fixation at proximal and distal eccentricities), in each assessment were pseudorandomized within a block of 80 trials. The intertrial interval was set to 2.5 s to ensure the pupil diameter returned to baseline prior to the next trial [21].

Following the pre exercise oculomotor assessment, participants immediately commenced the exercise intervention, whereas the post exercise assessment began when participants’ heart rates were less than 100 beats per minute (i.e., <5 min following the cool-down). Each oculomotor assessment required less than 10 min to complete.

### 2.4. Data Reduction, Dependent Variables and Statistical Analysis

Gaze position data were filtered offline via a dual-pass Butterworth filter employing a low-pass cut-off frequency of 15 Hz. Filtered displacement data were used to calculate the instantaneous velocities via a five-point central-finite difference algorithm. Acceleration data were similarly obtained from the velocity. Saccade onset was determined when velocity and acceleration exceeded 30°/s and 8000°/s, respectively. Saccade offset was marked by a velocity of less than 30°/s for 42 consecutive frames (i.e., 42 ms). Trials with missing data (i.e., loss of signal >25% of fixation period), RT less than 85 ms, and/or an amplitude less than 2° or greater than 26° were excluded from the data analysis (<10% of trials). 

Pupil data were filtered offline via a 10 Hz low-pass filter. Trials missing more than 40% of data or an eye position deviation more than 2° from the fixation during the initial fixation period (i.e., 0–1200 ms after fixation cross onset) were excluded from the analyses. A blink correction algorithm involving linear interpolation beginning 50 ms before the blink and ending 150 ms after the blink was used to avoid task-uncorrelated high-frequency changes in pupil size [22]. A pupil size greater than 2.5 standard deviations from a participant’s mean were also removed (<15% of trials). Notably, for all participants, at least 76% of trials were available for the statistical analyses. At least 20 trials remained for each condition from each participant. Because video-based tracking systems can distort pupil size following changes in gaze location, this measure was restricted to epochs involving central fixation and prior to saccade initiation (i.e., when gaze was located at the center of the screen). In line with previous work, [14,23,24] pupil size was determined in three epochs prior to saccade initiation (i.e., when gaze was located at the center of the screen): (1) the start of the visual fixation (FIX_st_; 100–300 ms after fixation onset), (2) maximal pupil constriction (CON_max_; 650–750 ms after fixation onset), and (3) end of gap (GAP_end_; 150–200 ms following gap onset; Figure 1).

Dependent variables included the reaction time (RT; time from response cueing to saccade onset), saccade duration (time from saccade onset to saccade offset), percentage of directional errors (i.e., the percentage of trials involving a prosaccade instead of instructed antisaccade and vice versa), baseline pupil diameter (average pupil diameter during FIX_st_), and task evoked pupil dilation (TEPD; GAP_end_ minus CON_max_). Dependent variables were analyzed via two (assessment: pre- and post-exercise) by two (task: pro- and anti-saccade) fully repeated measures ANOVA (*p* < 0.05). An alpha level of 0.05 was used for statistical significance, and simple-effects (i.e., paired-samples t-tests) were employed to decompose the main effects and interactions.

## 3. Results

The RT results yielded a main effect for the assessment (*F*(1,15) = 8.70, *p* = 0.01, and η_p_^2^ = 0.37) and task (*F*(1,15) = 34.9, *p* < 0.01, and η_p_^2^ = 0.7). The prosaccade RTs (260 ms, SD = 42) were shorter than the antisaccades (302 ms, SD = 36; Figure 2A), and the difference scores in Figure 2B show that RTs for the pro- and anti-saccades decreased from the pre- to post-exercise assessments. Directional errors also demonstrated a main effect of task (*F*(1,15) = 6.07, *p* = 0.026, and η_p_^2^ = 0.29), such that the prosaccades produced fewer directional errors (4%, SD = 4) than the antisaccades (9%, SD = 6; Figure 2C). Difference scores in Figure 2D show that this result did not vary from pre- to post-exercise assessments: (*F*(1,15) = 0.06, *p* = 0.804, and η_p_^2^ < 0.01). In terms of the saccade duration, the grand mean was 55 ms (SD = 0.6) and no reliable main effects or interactions were observed (all *F*(1,15) < 0.47, *ps* > 0.5, and all η_p_^2^ < 0.03).

Baseline pupil diameter produced a main effect of assessment (*F*(1,15) = 11.95, *p* = 0.004, and η_p_^2^ = 0.44), whereas TEPD demonstrated the main effects for the assessment (*F*(1,15) = 5.97, *p* = 0.027, and η_p_^2^ = 0.29) and task (*F*(1,15) = 4.58, *p* = 0.049, and η_p_^2^ = 0.23). Figure 3C,E show that baseline pupil diameter and TEPD decreased and increased, respectively, from pre- to postexercise assessments. As expected, the baseline pupil diameter did not significantly differ between the pro- and antisaccades (Figure 3A), while the TEPDs for the prosaccades were less than TEPDs for the antisaccades (Figure 3B). 

## 4. Discussion

The pro- and antisaccade RTs decreased from pre- to postexercise assessments. In terms of the antisaccade findings, the decrease in RT was independent of any change in saccade duration or directional errors—a result evincing that improved planning times were unrelated to a speed−accuracy trade-off. These results support previous work by our group [12,25], and provide convergent evidence that a single bout of aerobic exercise improves executive function. It is, however, important to recognize that prosaccade RTs also decreased from pre- to postexercise—a finding not observed in previous work. In reconciling this discrepancy, we note that previous work examined pro- and anti-saccades in separate blocks, whereas the current work randomly interleaved task-type on a trial-by-trial basis—a necessary manipulation to prevent TEPD attenuation due to task predictability [26]. As such, the current paradigm not only required the executive demand of response suppression for antisaccade trials, but also the executive component of cognitive flexibility (i.e., task-switching) across antisaccades and prosaccades [13,23,27]. Furthermore, and given that cognitive flexibility and task-switching efficiency elicits a robust postexercise benefit [4,28], it is possible that the postexercise decrease in prosaccade RTs observed here reflects a global benefit to executive function. 

Wang and colleagues [14] established that interleaved pro- and anti-saccades produce equivalent baseline pupil sizes; however, the latter task-type was associated with larger TEPDs. The increased TEPDs for antisaccades, in combination with single-cell recording work in non-human primates [29,30], has been taken to evince that TEPDs are a direct neural proxy for the increased executive demands of the antisaccade task (i.e., response suppression). In support of Wang et al. [14], we found that pre- and postexercise TEPDs were larger for antisaccades than prosaccades. Of course, the goal of the present work was to extend Wang et al.’s results in determining whether an exercise intervention modulates pro- and antisaccade TEPDs. To that end, our results demonstrate that baseline pupil size and TEPDs decreased and increased, respectively, from pre- to post-exercise. In accounting for this, Aston-Jones and Cohen [7] proposed that the modulation of activity in the LC-NE system underlies an optimal range of arousal (i.e., tonic activity) that serves to enhance neural gain (i.e., phasic activity) in executive-related cortical structures. Specifically, the activation pattern of the LC-NE system exhibits a causal relationship with behavioral performance and attention. Indeed, microinjection experiments in non-human primates have demonstrated that increasing the tonic activation of the LC-NE system via injection of a muscarinic cholinergic agonist increases distractibility, reduces phasic responsiveness, and decreases performance [7]. In contrast, suppressing tonic activation to moderate levels via the injection of an adrenoreceptor agonist decreases distractibility, increases phasic responsiveness, and increases performance. Bouret and Sara [6] proposed that these moderate levels of tonic activity in the LC entrains other neural systems to reduce responsiveness to irrelevant stimuli, thus preventing distractions, with the task-related phasic bursts of activity serving to selectively facilitate goal-directed behaviors by providing a brief attentional filter. Accordingly, we propose that the postexercise decrease in pro- and anti-saccade RTs, combined with the decrease in preparatory phase tonic pupil size, reflect an optimal modulation of the LC-NE system. Importantly, this modulation is proposed to underly enhanced attentional control via the processing of task-relevant information and increased phasic recruitment of executive control networks supporting saccade generation. 

An alternate account for the observed pupillometry findings is that the increased TEPDs across both pro- and anti-saccades reflect a broader improvement to general cognition. Our paradigm employed an interleaved pro- and antisaccade condition that introduced the executive component of cognitive flexibility. Our paradigm employed an interleaved pro- and antisaccade condition that introduced the executive component of cognitive flexibility. In line with the RT findings, the current prosaccade TEPD results provide evidence of a postexercise benefit to cognitive flexibility and task-switching efficiency. In support of this, previous work by our lab involving a blocked pro- and antisaccade paradigm with the same target eccentricities demonstrated a selective postexercise antisaccade benefit [13,23,27]. Additionally, another study involving concussed individuals demonstrated evidence of suppressed pro- and antisaccade RTs in a similar interleaved saccade paradigm that were proposed to reflect a concussion-related dysfunction to inhibitory control and cognitive flexibility [23]. Taken together, this evidence supports the stance that the current findings reflect a global improvement to executive function rather than a broader improvement to general cognition.

We note that our work contradicts the postexercise pupillometry results reported in a similar study by McGowan and colleagues [31]. This work examined the behavioral, electrophysiological, and pupillometric changes in response to the Eriksen flanker task before and after a 20-min bout of either aerobic exercise (i.e., 70% of the age-predicted maximum heart rate) or an active-control condition (i.e., walking on a treadmill at the lowest speed (0.5 mph) and incline (0 settings)). The discrepant findings may be accounted for by several between-experiment differences in methodology. First, the computation of TEPD in the current study was based on a baseline correction with CON_max_ (i.e., maximal pupil contraction)—a necessary procedure to control for pupil size variability in response to the pupillary light reflex associated with stimulus presentation [14,32]. In contrast, McGowan et al. employed a baseline correction for the initial size of the pupil prior to stimulus presentation. Although McGowan et al.’s procedure provides a normalization of pupil size change with respect to stimulus/task onset, it does not account for the additional variability that an individual’s pupillary light reflex contributes to pupil size once a TEPD response emerges [14,32]. Second, to maintain an accurate measure of the pupil size, the selected epochs for pupil analysis in the current study were either during the central fixation period or before saccade initiation; that is, when the eye position was located at the center of the screen and no other motor responses were being completed. In contrast, McGowan et al. selected a pupil epoch that included a manual motor response associated with the task, and it is unknown if the participants’ eye positions were fixated and directed to the center of the screen during this epoch. Finally, McGowan and colleagues employed the letter versions of the Eriksen flanker task, whereas the current study employed an interleaved pro- and antisaccade task. The benefit of the saccade paradigm used here is that it provides a directed measure of executive function without introducing concurrent non-executive task processes such as receptive language (i.e., letter identification), and does not result in larger manual-motor movements from impacting pupil responses and measurement [22]. 

### Study Limitations

We recognize that our findings are limited by at least three methodological factors. First, the current study did not employ a non-exercise control condition. As a result, we cannot directly assert that the post exercise changes in saccade performance and pupillometry metrics are specific to exercise or underlie a practice-related improvement in the current task. With that being said, our lab has repeatedly shown that a non-exercise control condition does not exhibit a practice-related improvement in pro- or antisaccade performance measures when performed in separate blocks [11,12] or randomly interleaved trials [4,23,28]. Specifically, three studies by our group [12,25,28], employing null hypothesis testing in conditions involving exercise (same exercise intensity used in the current study) and control (rest) conditions, reported that antisaccade RTs reliably decreased from to pre- to postexercise (all ps < 0.001; all d_z_ > 1.10), whereas no reliable change was associated with the pre- to post-rest assessments (all ps > 0.50, all d_z_ < 0.14). Moreover, here, we computed supplementary two one-sided test (TOST) statistics from our group’s previous work, and the results showed that pre- to post-rest antisaccade RTs for Samani and Heath (t(24) = 1.79, *p* = 0.043) [12], Heath and Shukla (t(17) = 2.07, *p* = *0*.027) [28], and Tari et al. (t(14) = 1.88, *p* < *0*.044) [25] were all within an equivalence boundary. Accordingly, null and equivalence tests from previous work support the direct assertion that antisaccade performance metrics do not relate to a practice-related improvement; rather, the results indicate that improved antisaccade RTs are specific to an exercise intervention. As such, we believe that the current findings, in combination with the extant literature, support the view that the oculomotor changes reported here reflect the exercise intervention. Second, the participants were young and reported a healthy lifestyle as determined via the PAR-Q and GLTEQ. It is therefore unknown whether populations outside this age range and fitness level would demonstrate a comparable postexercise improvement in arousal and executive function. Lastly, the postexercise assessment was completed within 15-min of the exercise intervention. Therefore, it is unclear whether the executive “boost” associated with LC-NE system modulation persisted beyond 15 min. In a follow-up study, we will examine pro- and antisaccade performance and pupillometry metrics across a range of post exercise intervals (i.e., immediate, for 30, 45, and 60 min) to determine the time frame through which a single bout of exercise modulates LC-NE activity and supports a postexercise benefit to executive function.

## 5. Conclusions

The present findings demonstrate that a 20-min single-bout of moderate-intensity aerobic exercise improved task-related arousal and preparatory activity in pro- and antisaccades. Specifically, the results demonstrated a pre- to postexercise decrease in pro- and antisaccade RTs, decreased tonic baseline pupil size, and increased phasic TEPDs. The results demonstrate that exercise-related saccade performance improvements are associated with a decrease in cognitive/psychological arousal—closer to an optimal level for task engagement—and augmented phasic recruitment of executive control resources. Accordingly, our findings provide evidence to suggest that the modulation of the LC-NE system is a mechanism underlying exercise-induced enhancements in cognition. 

## Figures and Tables

**Figure 1 brainsci-11-01048-f001:**
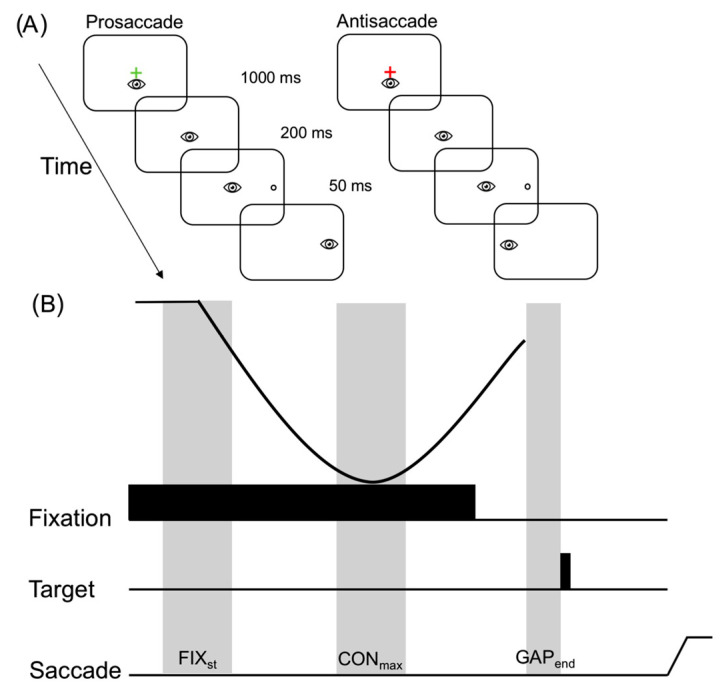
Panel A shows the timeline of visual and motor events for pro- and anti-saccades. A single target eccentricity is depicted to the right of the fixation; however, in the current study, two eccentricities were employed and the targets are presented left and right of the fixation. Panel B presents the epochs for the pupil analysis: fixation start (FIX_st_; 100–300 ms after fixation onset), maximal pupil constriction (CON_max_; 650–750 ms after fixation onset), and gap end (GAP_end_; 150–200 ms after gap onset). The solid black line in Panel B depicts the time course change in the absolute pupil diameter.

**Figure 2 brainsci-11-01048-f002:**
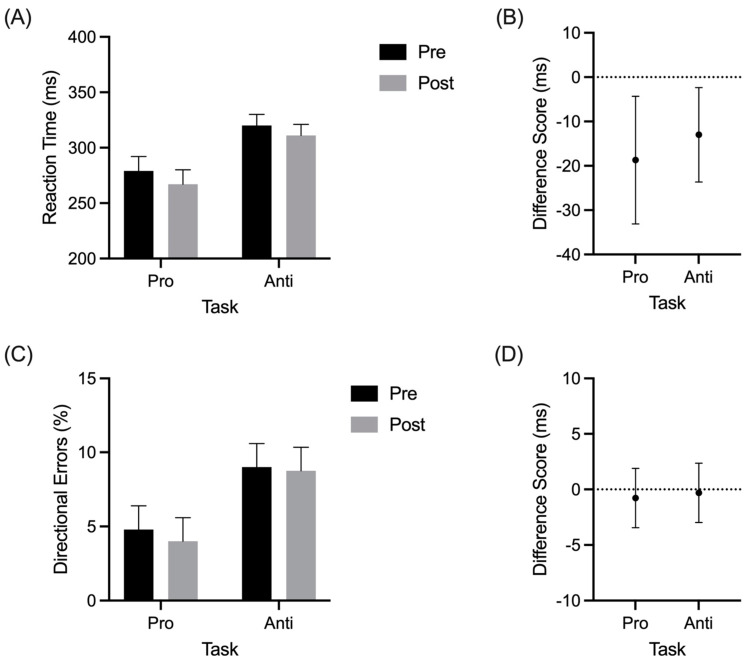
Group mean reaction time (RT) (**A**) and directional errors (**C**) for pro- and antisaccades at pre- and postexercise assessments. The offset panels show an RT (**B**) and directional error (**D**) difference scores (postexercise minus pre-exercise) with error bars representing 95% between-participant confidence intervals (**B**). An absence of overlap between the error bars and zero (i.e., horizontal dashed line) indicates a reliable difference inclusive to a test of the null hypothesis.

**Figure 3 brainsci-11-01048-f003:**
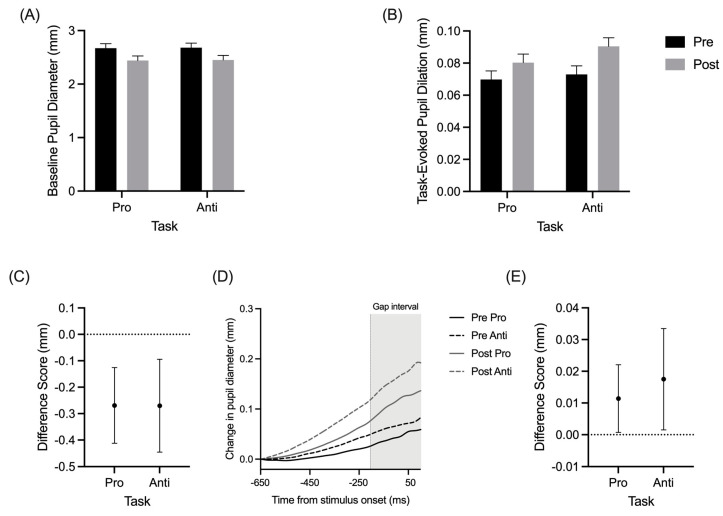
Group mean absolute pupil diameter (**A**) and task-evoked pupil dilation (**B**) for pro- and anti-saccades at pre- and postexercise assessments. Group mean difference scores (postexercise minus pre-exercise) for pupil diameter (**C**) and task-evoked pupil dilation (**E**). Baseline corrected pro- (solid lines) and antisaccade (dashed lines) pupil size changes by time traces for an exemplar participant during pre- (black lines) and postexercise (grey lines) assessments (**D**). Error bars represent 95% between-participant confidence intervals.

## Data Availability

Data are available from the corresponding author upon request. The data are not publicly available due to ethical restrictions.

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
