# Peer review of "Pupillometry Reveals the Role of Arousal in a Postexercise Benefit to Executive Function"

_brainsci, 2021, doi:10.3390/brainsci11081048_

Round 1

Reviewer 1 Report

This paper examines the effects of 20-min exercise on the performance in the interleaved pro- and anti-saccade task to investigate the effects of aerobic exercise on executive function. The authors find that in addition to the typical anti-saccade effects on SRTs (saccade reaction time) and direction error rates, SRTs in both pro- and anti-saccade conditions were faster in the post-, compared to, the pre-exercise condition. But, no interaction was obtained. Similar effects were observed in direction errors. Moreover, baseline pupil diameter (tonic) was smaller and task-evoked pupil dilation was larger in the post-, compared to, pre-exercise condition. The authors argue that the observed improvement in saccade behavior is attributed to exercise-induced changes in arousal, arguably mediated by the LC, that can improve executive function. Overall, I think the results are novel and very interesting, but there are some issues that need to be addressed.

The authors argue that arousal plays a critical role in mediating this exercise-induced improvement, and “if exercise enhances LC-NE system attentional modulation via an arousal mechanism then decreased postexercise antisaccade RTs should be paired with a pre- to postexercise decrease in baseline pupil size (i.e., suppression of tonic activity to moderate levels) and a concomitant increase in TEPD”. This rationale seems not clear, because many studies demonstrate that the higher arousal level (induced by emotional stimuli or by cognitive demanding processes) correlates with larger pupil diameter, with a higher level of LC activity (see Aston-Jones 2005 or Gilzenrat et al., 2010 cited in line 57-59). Since the authors found that baseline pupil diameter “decreases” after exercise, do the authors argue that arousal level decreases after exercise? If this is the case, this is counterintuitive, as it is assumed that arousal level should be increased after exercise. The current results are indeed interesting, but the hypothesis and rationale need to be more clearly articulated. First, how does "arousal level" modulate tonic and phasic pupil size? Then, how arousal level could influence behavioral performance? And if their results are different from the predictions, they should be clearly discussed and provide some reasons in the discussion.

The aim of the study is to investigate the arousal mechanisms underlying exercise improvement on executive function. However, the results only show the main effect of exercise on SRTs (or the error rate), not the interaction between pre/post and pro/anti. As argued in the discussion, it is understandable that pro-saccade behavior could also be improved from the general boost due to exercise. And task-switching is involved in the interleaved pro- and anti-saccade task, thus there is a bit of “executive” control involved in the pro-saccade condition. Still, the absence of greater anti-saccade improvement suggests that the exercise improvement on “executive” function is not supported by the current results. Instead, the current results only reveal the general benefits after a short exercise. I think this needs to be clearly discussed in the manuscript. There may be some possibilities. First, the anti-saccade manipulation was not very effective in this study, as direction error rates were only 9% in the anti-saccade (4% for pro-saccade), suggesting that this task was pretty easy for all participants, as their correction rate was 91%. I think this could be due to the long distance between the target and the fixation point (13.5 and 16.5 deg). Second, if I understand correctly, there are only 80 trials in each experiment, so only 40 trials for the pro- or anti-saccade condition. Therefore, there may be a limited number of trials per condition after excluding bad trials, given there are only 16 subjects, so statistics power may be too weak to reveal the interaction. Alternatively, these results suggest that exercise does not specifically boost executive function.

Minor points:

  1. Having a direction error figure is also helpful for the readers to capture the main part of the results, as SRT and direction errors are the major effects in the pro- and anti-saccade task. Lastly, the texts can be utilized better to clearly describe each “panel” figure (e.g., Fig. 2A, 2B, Fig. 3A…).
  2. Any reason for using two target distances (13.5 and 16.5 deg), but not separated these two distances for analysis?
  3. After exclusion, how many trials are left per condition should be reported.

Reviewer 2 Report

In this work, the authors propose that 20 minutes of physical exercise can affect some oculomotor measures. One single experiment is reported, with a rather limited sample size (N = 16).

I’m aware that the covid pandemic is limiting lab-based research extensively, but I’m a bit confused about this work. More precisely, I feel that in the current form this paper prevent any conclusion to be drawn because what is missing here is a sort of “control” condition in which participants do not complete the 20-min gym session (I think they might just relax). Otherwise, I think we are not allowed to say anything about the effects, if any, of physical exercise on eye movements; the simplest conclusion is that the results reported here are merely due to learning/experience-based mechanisms.

Round 2

Reviewer 2 Report

The paper is fine.